# Comparative Transcriptome Analysis Reveals Genetic Mechanisms of Sugarcane Aphid Resistance in Grain Sorghum

**DOI:** 10.3390/ijms22137129

**Published:** 2021-07-01

**Authors:** Desalegn D. Serba, Xiaoxi Meng, James Schnable, Elfadil Bashir, J. P. Michaud, P. V. Vara Prasad, Ramasamy Perumal

**Affiliations:** 1United States Department of Agriculture—Agricultural Research Service, U.S. Arid Land Agricultural Research Center, Maricopa, AZ 85138, USA; des.serba@usda.gov; 2Department of Agronomy and Horticulture, University of Nebraska, Lincoln, NE 68588, USA; meng0170@umn.edu (X.M.); schnable@unl.edu (J.S.); 3Department of Agronomy, Kansas State University, Manhattan, KS 66506, USA; ebashir@ksu.edu (E.B.); vara@ksu.edu (P.V.V.P.); 4Department of Entomology, Kansas State University, Hays, KS 67601, USA; jpmi@ksu.edu; 5Agricultural Research Center, Hays, KS 67601, USA

**Keywords:** sorghum, sugarcane aphid, RNA-seq, transcriptomics, gene expression

## Abstract

The sugarcane aphid, *Melanaphis sacchari* (Zehntner) (Hemiptera: Aphididae) (SCA), has become a major pest of grain sorghum since its appearance in the USA. Several grain sorghum parental lines are moderately resistant to the SCA. However, the molecular and genetic mechanisms underlying this resistance are poorly understood, which has constrained breeding for improved resistance. RNA-Seq was used to conduct transcriptomics analysis on a moderately resistant genotype (TAM428) and a susceptible genotype (Tx2737) to elucidate the molecular mechanisms underlying resistance. Differential expression analysis revealed differences in transcriptomic profile between the two genotypes at multiple time points after infestation by SCA. Six gene clusters had differential expression during SCA infestation. Gene ontology enrichment and cluster analysis of genes differentially expressed after SCA infestation revealed consistent upregulation of genes controlling protein and lipid binding, cellular catabolic processes, transcription initiation, and autophagy in the resistant genotype. Genes regulating responses to external stimuli and stress, cell communication, and transferase activities, were all upregulated in later stages of infestation. On the other hand, expression of genes controlling cell cycle and nuclear division were reduced after SCA infestation in the resistant genotype. These results indicate that different classes of genes, including stress response genes and transcription factors, are responsible for countering the physiological effects of SCA infestation in resistant sorghum plants.

## 1. Introduction

Sorghum [*Sorghum bicolor* (L.) Moench] is a drought tolerant C_4_ grass species, which is grown for grain, forage, sugar, and biofuel. Sorghum is the fifth most important cereal crop in the world after wheat (*Triticum aestivum* L.), rice (*Oryza sativa* L.), maize (*Zea mays* L.), and barley (*Hordium vulgare* L.) [1,2]. Sorghum germplasm lines have broad genetic variation in economically important traits affecting productivity and resistance to biotic and abiotic stresses [3], but most genetic improvement programs have relied upon classical breeding approaches. Traits determining biotic and abiotic stress tolerance remain among the most challenging to improve, largely due to trait complexity, quantitative genetic contributions, and enormous interaction effects between genotype, environment, and cultural management. The recent invasion of North America by a sorghum-feeding strain of the sugarcane aphid, *Melanaphis sacchari* (Zehntner) (Hemiptera: Aphididae) (SCA), created new pest management challenges for sorghum producers [4,5]. Modern genomics approaches hold the potential to manipulate available genetic variation for improved crop resistance to the pest in a shorter time than conventional breeding techniques.

Sorghum has become a botanical model for plant genomics research on C4 grasses, largely because of its international economic importance [6]. It has a relatively small diploid genome (~750 Mbp) and can serve as a useful genetic model for other tropical grasses with larger, more complex, genomes. The first sorghum genome sequence was based on a whole-genome shotgun sequence that assembled 730 Mbps, which were validated by genetic, physical, and syntenic information [7]. This analysis placed about 98% of genes in a chromosomal framework and revealed significant similarity to rice with respect to gene order and density. The newly sequenced genome of sweet sorghum has revealed a high level of genomic similarity to grain sorghum, irrespective of significant phenotypic differences [8].

The SCA has historically infested only sugarcane in the southern United States [9,10], but began attacking grain sorghum in northeastern Mexico, Louisiana, Texas, southern Oklahoma, and eastern Mississippi in 2013 [4]. Over the next few years, it became a major pest of sorghum wherever it is grown in North America [11,12,13]. In 2014 and 2015, sorghum growers in the lower Rio Grande Valley of south Texas lost more than US $31 million due to reduced yields and the cost of insecticide applications [14]. The aphid has now been confirmed in all 18 sorghum-growing states of the USA [15] and only a few grain sorghum hybrids have expressed moderate resistance to the SCA [16]. The aphid has a high reproductive rate on susceptible cultivars [17], disperses far and wide with prevailing winds [15], and can overwinter on remnant sorghum and Johnsongrass, *Sorghum halapense*, as far north as the Oklahoma border with Texas [12,18]. Susceptible sorghum hybrids are still grown over large acreages, such that most sorghum-producing regions of the USA remain at risk. Although natural biological control has evolved rapidly to reduce peak SCA populations on the High Plains [19,20], recruitment of aphid predators and parasitoids to SCA-infested sorghum has, to date, been insufficient to preclude the need for chemical control elsewhere, such as on the east coast of USA [21], possibly because this region lacks the large acreage of winter wheat that is a major source of aphid natural enemies migrating to summer crops in central North America.

Historically, host-plant resistance has been a useful tactic for managing aphids in cereal crops [22], including sorghum [17,23,24]. Resistant sorghum cultivars and hybrids have the potential to provide the foundation of an environmentally sound, sustainable management program for SCA when employed in combination with other integrated pest management practices. Accordingly, sorghum breeding programs have identified some commercial sorghum hybrids and parental lines with partial resistance [11,25]. A better understanding of the genetic and molecular mechanisms underlying plant resistance to SCA would assist the redeployment of existing resistance genes into new lines and perhaps identify novel resistance genes. However, breeding for SCA resistance is hampered by a lack of understanding of the molecular bases of aphid virulence, host plant resistance mechanisms, and the genes involved. Identification of the genes involved in sorghum responses to SCA infestation would have practical value for plant breeding efforts to combat this invasive pest and could enhance our understanding of resistance mechanisms. The present study was conducted to achieve three major aims: (i) to characterize the genes that are differentially expressed between SCA-resistant (TAM428) and susceptible (Tx2737) sorghum genotypes; (ii) to elucidate patterns of temporal change in the expression of genes during SCA infestation; and (iii) to identify sequence-specific DNA-binding factors (transcription factors) that control the transcription of genes differentially expressed in response to SCA feeding.

## 2. Results

### 2.1. RNA-Seq Data Summary

To assess the global transcriptome profile of sorghum in response to SCA infestation, we performed RNA-Seq analysis on resistant (R, TAM428) and susceptible (S; Tx2737) sorghum genotypes. Paired-end sequencing of 36 RNA-Seq libraries on Illumina HiSeq X Ten System generated an average of 21,358,630 reads per sample (Appendix A). The sequence trimming and quality control processes retained an average of 96% of the reads for all 36 samples. On average, 20,450,495 reads per sample (95.75%) were found with good quality for further analysis. The mapping rate of sequenced libraries ranged from 86.7 to 98.3% with an average of 95.5% (Appendix A).

### 2.2. Differentially Expressed Genes (DEGs)

Characterization of DEGs between the R and S genotypes over time following SCA infestation represents a first step in gaining biological insight into the mechanisms of SCA resistance in sorghum. A total of 18 predetermined comparisons among the samples were performed between R and S lines within and among time points (Figure 1). These differential gene expression analyses were conceptualized prior to conducting the experiment to characterize the inherent differences between the resistant and susceptible genotypes and to identify the dynamics of altered gene expression over time after SCA infestation (Appendix A). The number of differentially expressed genes differed dramatically between the R and S genotypes within and across time points for the same genotype, as compared with untreated controls.

A trend toward increasingly differential gene expression was observed between resistant and susceptible genotypes within time points after SCA infestation, the largest difference being observed at 360 h (Appendix A). The fewest genes were differentially expressed between R96 and RC0, and S96 and SC96 also showed few differential expressions. Few differential expressions imply that aphid feeding did not trigger significant responses in either R or S genotypes triggered significant responses to aphid feeding at 96 h after infestation.

The numbers of DEGs in the R genotype was generally low compared to the S genotype at specified time points after infestation. The greatest significant difference (*p* ≤ 0.05) was observed for S360 h vs. SC0 h, with 6794 differentially expressed genes at the log2 (foldchange). A plausible explanation for these differences is the effective elicitation of catabolic processes by aphid feeding on the S genotype that serve to improve nutritional quality of phloem sap for the SCA.

### 2.3. Dynamics of Differential Gene Expression in Response to SCA Infestation

The statistical significance (*p*-value) and fold change of gene expression between treated and control plants at 96 h and 216 h post-infestation, for both R and S genotypes revealed significant differences in gene expression between control and infested samples in gene expression (Figure 2). A larger number of genes of the R genotype were significantly upregulated in treatment samples than in untreated controls at both 96 h and 216 h post-infestation (Figure 2A,B). On the other hand, similar numbers of genes were upregulated in treated and untreated plants of the S genotype (Figure 2C,D), although a larger number of genes were expressed at 216 h.

### 2.4. Identification of Commonalities and Differences of DEGs

A Venn diagram (Figure 3) was constructed to visualize commonalities and differences in genes expressed in the R and S genotypes at specified time points. All upregulated transcripts at each time point were compared for commonality and difference with their counterparts in the other genotype at the same time (Figure 3A). At 0 h, the R and the S genotypes had 1637 and 2414 transcripts upregulated, respectively, whereas 580 transcripts were commonly expressed.

The time points R96 h, R216 h, and R360 h shared more upregulated genes than did their susceptible counterparts. In addition, the R360 h samples had more upregulated genes than did the S360 h samples (1338 vs. 781). At 0 h and 96 h, more genes were upregulated in the S genotype than in the R at the same time points. At 96 h post-infestation, a smaller overall number of transcripts were expressed (220 in the R and 1905 in the S, with 197 in common between the two genotypes). This stands in contrast to the 2179 and 3711 transcripts upregulated in the R216 h and S216 h samples, respectively, post-infestation, and the 1629 and 4353 transcripts upregulated in the R360 h and S360 h samples, respectively, post-infestation. There were also 796 transcripts upregulated in both genotypes at 360 h post infestation.

We also compared gene expression in un-infested controls of the R and S genotypes at different times (Figure 3B) and observed 1637, 220, and 578 upregulated genes in the R genotype at 0, 96, and 216 h, respectively. Similarly, 3216, 40, and 940 genes were upregulated in the S genotype at the same time points. In both cases, commonly expressed genes were higher between 0 h and 216 h than between 0 h and 96 h, indicating that responses to SCA feeding increase progressively with infestation time.

To assess gene expression patterns post-infestation, we compared the R and S genotypes on each of the four time points (Figure 3C). In general, there were more commonalities in expressed genes across time points for the S than for the R genotype. Overall, only 40 genes were consistently upregulated in the R, compared to 268 in the S genotype. Thus, the pattern of gene expression was more dynamic in the R than in the S genotype, likely reflecting the ontogeny of defensive responses by the R line over the course of aphid infestation.

### 2.5. Annotation and Clustering of Expressed Genes

Cluster and gene ontogeny (GO) enrichment analyses revealed different gene response patterns at different times after SCA infestation (Figure 4). Six clusters of gene expressions were deduced by heat map and graphical representation. Expression of genes regulating protein and lipid binding, cellular catabolic processes, transcription initiation, and autophagic processes began at 96 h post-infestation and steadily increased from 216 h to 360 h (Cluster 1). Similarly, expression of genes underlying responses to external stimuli, starvation and stress, cellular communication, UDP glycosyltransferase activity, and proton transport began at 96 h, and showed sharp increases after 216 h (Cluster 3). In contrast, the expression of genes related to catalytic activity, oxidoreductases, acyl and glycosyl transferases, hydrolases, electron transport activities, phosphorylation, and protein folding, steadily decreased over time post-infestation (Cluster 2). Likewise, genes responsible for developmental processes such as DNA binding, regulation of cellular division and cell cycle, DNA metabolic processes, and membrane-bound organelles diminished with time post-infestation (Cluster 4). These genes were highly expressed at time of infestation (0 h) and their expression progressively diminished over time post-infestation.

### 2.6. Gene Ontology Enrichment of DEGs

To comprehend the biological and molecular functions of the DEGs in relation to plant growth, development, and defense, GO enrichment analysis based on gene annotation databases was conducted. The results elucidated the fundamental molecular differences between R and S genotypes, as well as gene expression dynamics over the course of SCA infestation (Figure 5). At 0 h, genes modulating biological processes such as photosynthesis, generation of precursor metabolites and energy, metabolic processes, and cellular components such as the intracellular ribonucleoprotein complex, thylakoid, and ribosome were differentially upregulated in the R genotype with significant GO terms (Appendix A). At the same time (0 h post-infestation), most of the genes upregulated in the susceptible genotype had molecular functions such as catalytic activity, carbohydrate binding, transferase activity, nucleotide binding, and kinase activity. On the other hand, the commonly upregulated genes encoded regulation of biological processes, cellular processes, homeostasis, and extracellular components.

Differences in gene expression between the R and S genotypes were evident starting at 96 h after SCA infestation, including enrichment of genes encoding stress responses in the R genotype (Appendix A). This trend of enrichment continued, and at 216 h post-infestation, genes modulating metabolic processes, and responses to biotic stimuli and stress, were all upregulated in the resistant genotype (Figure 5).

### 2.7. Transcription Factor Expression in Response to SCA Infestation

The R and S sorghum genotypes differed in transcriptional expression over time post-infestation. Different families of transcription factors (TFs) such as EBER (Epstein-Barr virus-encoded small RNA), GRASS [gibberellic-acid insensitive (GAI), repressor of GAI (RGA) and scarecrow (SCR)], ARF (Auxin Response Factors), NAC (NAM/ATAF/CUC that refers to no Apical Meristem, NAM), ATAF (Arabidopsis Transcription Activation Factor), and CUC (Cup-shaped Cotyledon), MADS [Minichromosome Maintenance Factor (M), Agamous (A), and Deficiens (D), and Serum Response Factor (S)], bZIP (basic Leucine Zipper), bHLH (basic Helix-Loop-Helix), WRKY, and MYB were all implicated in major roles regulating plant responses to SCA infestation (Figure 6, Appendix A). Different families of TFs were up- or down-regulated at different time points. Five among seven ARF TFs, except for ARF12 and ARF24, were more highly expressed in the R than in the S genotype at 360 h post-infestation (Figure 6A). At 216 h post-infestation, un-infested susceptible controls exhibited upregulation of ARF24 and ARF26.

Nine MADS TFs were differentially expressed in the two genotypes at different time points post-infestation (Figure 6B). At 360 h post-infestation, MADS52, MADS18, MADS56, and MADS26 were upregulated more in the R than in the S genotype. On the other hand, the S genotype overexpressed MADS67, MADS22, and MADS2 at 360 h post-infestation when compared to the R genotype. Out of 33 NAC TFs differentially expressed, 18 were overexpressed in the S genotype (Figure 6C). In contrast, the R genotype consistently overexpressed NAC116 and NAC122 at 96 h post-infestation. Another NAC TF, NAC91 was overexpressed by the R genotype at 360 h, and by susceptible controls at 216 h and 96 h post-infestation. NAC4 and NAC20 were overexpressed in resistant controls at 0 h and 96 h post-infestation, and then remained mostly downregulated.

Most of the 18 GRAS TFs differentially expressed were silent during early infestation but became upregulated in un-infested resistant controls and the S genotype mostly at 216 h and 360 h post-infestation (Figure 6D). The exceptions to this were GRAS44, GRAS72, and GRAS73, which were highly expressed by the R genotype (both infested and un-infested) early in the course of infestation. There were 33 WRKY genes expressed in this study (Figure 6E). Except for WRKY18, WRKY50, WRKY70, WRKY73, and WRKY75, most of the genes regulating WRKY were also more upregulated in the S genotype than in the R genotype at 216 h and 360 h post-infestation. These five WRKY genes were also upregulated in the R genotype at 360 h post-infestation. Similarly, most of the EREB TFs were upregulated late in the infestation process, and more upregulations were observed in the susceptible than in the resistant genotype (Appendix A). However, EBER28, EBER30, EBER31, and EBER49 were more upregulated in the R than in the S genotype. Expression of the different GRAS and ARF TFs was also low in un-infested controls and early in the infestation process, whereas higher expression of GRAS in the S than in the R genotype was observed late during infestation.

## 3. Discussion

Plants have evolved various morphological and physiological mechanisms to respond to the stresses imposed by herbivory and counter their negative impacts on fitness [26,27]. Among herbivores, aphids in particular have evolved complex parasitic relationships with their host plants, often mediated by signaling compounds (‘elicitors’) that affect plant gene expression and metabolism in ways that can subvert normal plant defensive responses and improve the quality of phloem contents as food, usually at the expense of plant fitness [28,29,30]. The molecular basis of resistance to SCA in sorghum remains poorly understood, although a previous study on global transcription responses of sorghum genotypes resistant (RTx2783) and susceptible (A/BCK60) to SCA reported suppressed expression of multiple sugar- and starch-associated genes in resistant plants at five and 10 days post-infestation [31]. Such changes would be consistent with the aphid eliciting changes in plant physiology that raise phloem nitrogen content at the expense of carbohydrate content. The same study identified several nucleotide-binding sites, leucine-rich repeat (NBS-LRR) and putative aphid resistance genes [31].

We observed substantial differences in the numbers of DEGs between the R and S genotypes at different time points post-infestation. The reaction of R and S plants to the SCA likely differ in terms of the number, type, and timing of the gene expression changes as they respond to the physiological assault of the pest on processes associated with the normal growth and development of the plant. Indeed, differential expression analysis showed a trend toward increasing DEGs over the course of infestation in both R and S genotypes (Figure 5). This likely reflects cascades of host plant gene expression in response to feeding elicitors injected into the phloem sap with the ‘watery’ fraction of aphid saliva [32], with initial changes in gene expression triggering subsequent changes in the expression of others. Phloem elements carry assimilates from source to sink and are the site of many physiological changes induced by aphid feeding that result in attenuation of host defenses, and thus may also be the site of plant gene expression associated with phloem-based resistance [33]. Aphid feeding on poor quality or non-host plants is typically prevented because aphids are unable to access phloem elements [34], so resistance in acceptable host plants is typically expressed via physiological interactions that occur during aphid salivation and ingestion within the phloem. Kiani and Szczepaniec [13] reported transcriptional changes in response to *M. sacchari* feeding on resistant sorghum that were lower or absent in a susceptible counterpart, and noted these responses were more pronounced when feeding occurred on seedlings (2 weeks post-emergence) than on adult plants (6 weeks post-emergence). Upregulation of known plant defense genes in resistant plants were also noted, similar to our findings.

Genes enriched in the R genotype at 216 h and 360 h post-infestation included those that respond to biotic stimuli (GO:0009607), coordinate stress responses (GO:0006950), regulate metabolic processes (GO:0008152), and mediate developmental processes (GO: 0051704). These changes in gene expression likely mediate enzyme production or activity in R plants specifically in response to aphid feeding. Genes regulating metabolic processes were also overrepresented in the R sample at 216 h and 360 h post infestation. Because metabolic processes represent the sum of catabolic and anabolic processes required to obtain energy and produce cellular components, the over-representation of these genes suggests that R plants accelerate metabolic processes to counteract the negative effects of aphid feeding on plant growth and development. An acceleration of metabolism in response to insect feeding often leads to the production of defensive metabolites such as alkaloids, terpenes, and glucosinolates, the exact profile of which depends on the plant species [35]. Pathway analysis of upregulated defense related genes in SCA-resistant sorghum has identified hormone-signaling pathways, pathways coding for secondary metabolites, glutathione metabolism, and plant-pathogen interaction [13].

At 360 h post infestation, genes modulating cellular protein modification processes (GO:0006464) and macromolecule assembly (GO:0043412) were upregulated in the R line, relative to the S line. Protein modifications can involve pre-translational, co-translational, and post-translational alterations of amino acids in proteins, peptides, and nascent polypeptides which can change their stability or functionality, with effects on plant signaling, gene expression, and enzyme kinetics [36,37]. Thus, the R genotype modified its enzyme functionality, signaling, and gene expression in response to aphid feeding in ways that the S genotype did not. Genes encoding other molecular functions such as carbohydrate binding (GO: 0030246) were also overrepresented in the R sample at 216 h post infestation. Carbohydrate-binding proteins such as lectins have anti-herbivore functions in many plant species [38], and are known to have antibiotic activity against aphids [39]. Their enrichment in the R line when infested with SCA suggests that carbohydrate-binding proteins may play a role in SCA resistance.

Genes controlling the activity of kinases (GO:0016301) and transferases with phosphorus containing groups (GO:0016740, GO:0016772) were overrepresented in the R genotype at 360 h. Kinases play a crucial role in phosphorylation and energy transfer. In the context of host plant resistance to insects, kinases and transferases are known to regulate the dynamics of phytohormones such as jasmonic acid (JA), ethylene, and salicylic acid (SA) in response to herbivory. They are also known to activate the transcription of herbivore defense-related genes, and mediate the accumulation of defensive metabolites [40].

Transcription factors (TFs) include a wide range of proteins involved in the initiation and regulation of gene transcription [41]. They are key components of plant regulatory networks that respond to environmental cues and mediate stress responses at the cellular level. Analysis of TFs produced in response to SCA herbivory revealed dynamic transcriptional changes in both S and R genotypes, suggesting that these genes are involved in general sorghum responses to aphid feeding, rather than mediating specific resistance traits. For example, Kariyat et al. [42] showed that an MYB transcription factor was associated with the accumulation of flavonoids in sorghum plants that conferred resistance to the corn leaf aphid, *Rhopalosiphum maidis* Fitch. However, several classes of TFs related to host-plant defense were overrepresented in our R sample compared to annotated genes of the sorghum genome [43].

In higher plants, there are a number of WRKY genes that have positive or negative effects on the transcription of defense-related genes [44]. Zheng et al. [45] reported that an overexpression of WRKY33 had a positive effect on regulation of JA-induced defense genes and a negative effect on regulation of SA-related defense genes. Both the SA [46] and JA [47] pathways have been implicated in responses to aphid feeding. We observed that more WRKY genes, including WRKY33, were upregulated in the S genotype than in the R genotype at 360 h, suggesting that these genes were responding to aphid elicitors that target plant physiology to benefit the aphid, and that suppression of this upregulation may be one component of resistance [48]. On the other hand, down-regulation of SA-related genes in the S genotype likely reflects mitigation of plant defensive responses by the aphid (e.g., [49]).

The NAC gene group (NAM, ATAF, and CUC) is one of the largest plant-specific TF families involved in plant development, stress responses [50] and plant immune responses [51]. Similarly, members of the MADS-box gene family are involved in developmental control, signal transduction, and stress responses in various plants [52,53]. Because TFs function to regulate the expression of multiple target genes, their loss or gain of function can lead to dramatic phenotypic alterations. The observed over-expression of several NAC and MADS-box gene families in the SCA resistant genotype would suggest they are involved in countering the physiological impacts of aphid feeding. It is not clear if immune receptor-mediated mechanisms mediate responses to SCA feeding in sorghum as they do in the *Arabidopsis*—*Myzus persicae* system [28]. However, susceptible plant responses to aphid feeding have been shown to involve the downregulation of key genes involved in plant immunity, as in zucchini plants responding to feeding by *Aphis gossypii* (Glover) [54].

The MYB TF is another large family of TFs involved in controlling plant growth and development, cell morphology, primary and secondary metabolic reactions, and stress responses [55,56]. The NAC TF has an evolutionary relationship with WRKY, as their DNA-binding domains interact with the same core sequence in target genes [57]. An overexpression of NAC TFs has been shown to improve biotic and abiotic stress tolerance in plants [58]. The MYB genes that are upregulated in the R genotypes, such as MYBR66, MYB66, MYBR40, and MYB92, have roles in transcription activation, zinc-ion binding, and protein–protein interactions [59], suggesting that these TFs could function to integrate multiple inputs to coordinate transcription of defense related genes [60].

Changes in plant gene expression in response to aphid feeding are, of course, only one side of the HPR coin, the other being changes in gene expression that occur in the aphid in response to specific plant resistance factors [61,62]. Additional work is warranted to examine the changes in SCA gene expression that occur in response to SCA feeding on the two sorghum lines used in this study. Taken together with the current findings, the results would permit the elucidation of reciprocal genetic responses in the aphid-plant interaction as it occurs in susceptible and resistant sorghum lines.

Although aphids have longstanding importance as cosmopolitan pests, genomic studies of aphids are lagging. Recently, the genome of the pea aphid was sequenced as a primary aphid model system [63]. In this study, RNA-Seq data collected from three replications were analyzed. Based on the analysis of variance, the respective *p* values and the level of significance are presented for all the differentially expressed genes (DEGs) for the biotic stress resistance—some of them are also known for insect resistance in the resistant genotype. The results developed through the robust and highly reliable next generation sequencing (NGS) data analysis in this study itself justifies validation for the RNA-Seq analysis. This initial genome sequencing may serve as a springboard for functional genomics and biological studies unique to aphids [64] and may facilitate a better understanding of SCA genome features that are key to its pest status on grain and forage sorghums.

## 4. Materials and Methods

### 4.1. Plant Materials

Two sorghum inbred lines, namely TAM428 (SCA resistant, R) and Tx2737 (SCA susceptible, S), were obtained from Dr. Chad Hays, USDA Lubbock, Texas, for use in the study. TAM428-SC110-9 was released in 1974 by Texas AgriLife Research Sorghum Improvement Program as a disease resistant inbred parental line. TAM428 is a parental line, a selection from the BC_3_F_2_ of the conversion of IS 12610 [65]. RTx2737 was released in 1976 (Tx7000–Tx2536 derivative) [66]. It is a parental line and green bug differential, resistant to biotype C [11].

### 4.2. Aphid Infestation and Sampling

The seeds of both genotypes were planted in plastic cones (25 × 5 cm) in a greenhouse and transferred to a climate-controlled growth chamber at the two-leaf stage at Agricultural Research Center-Hays, Kansas State University. The climate-controlled growth chamber was set to 23 ± 1 °C and a 14:10 h (light:dark) daylength, conditions known to facilitate both plant growth and SCA survival and reproduction [20] and plant defense mechanisms [67]. Seedlings of both the R and S genotypes were each infested at the three-leaf stage with five-fourth instar SCA nymphs. Each cone was then tightly covered with a tubular plastic cage, ventilated via perforations covered with organdy fabric, to confine the aphids within each replicate. Control plants were mock infested and held under the same conditions. After infestation, the plants were returned to the growth chamber under the same environmental conditions and aphid survival was checked after 24 h; any dead aphids were replaced to ensure that five aphids molted to apterous adults and began reproduction in each cone. The experiment was laid out in a randomized complete block design (RCBD) with three replications.

Leaf tissue samples were collected individually from both infested and un-infested plants of both sorghum genotypes at 0 h, 96 h, 216, and 360 h after SCA infestation. Samples were collected at exactly 10:00 am to control for diurnal cycles of transcription. Samples of three seedlings from each biological replicate in each treatment were pooled and transferred to −80 °C freezer where they were stored until RNA extraction. Sampling times were selected based on a preliminary observation that indicated when visible damage became evident on susceptible plants and how long they were able to survive infestation. Time gaps between samples were selected to be wide enough to detect changes in gene expression patterns in both the resistant and susceptible lines.

### 4.3. RNA Extraction, Library Construction and Sequencing

The total RNA in each sample was extracted from the leaf tissues samples using TRIzol^®^ reagent following the manufacturer’s instructions (Invitrogen, Carlsbad, California, USA), and genomic DNA was removed using DNase I (TaKaRa Bio, Mountain View, CA, USA). RNA quality was checked by Agilent bioanalyzer 2100 (Agilent Technologies, Santa Clara, CA, USA), and Nanodrop 2000 spectrophotometer (Nanodrop Technologies, Wilmington, DE, USA) was used for RNA quantification. After performing quality control (QC), the samples were outsourced to Psomagen (https://psomagen.com/dna-sequencing-services accessed on 23 October 2020) for library construction and sequencing. RNA-seq libraries construction of the qualified samples was conducted following the TruSeq^TM^ stranded mRNA sample preparation kit (Illumina, San Diego, CA, USA). A sequencing library was prepared by random fragmentation of the cDNA samples, followed by 5′ and 3′ Illumina TruSeq3 adapter ligation. Adapter-ligated fragments were PCR-amplified and gel-purified before being loaded onto a flow cell where fragments were captured on a lawn of surface-bound oligos complementary to the library adapters. Each fragment was then amplified into distinct, clonal clusters through bridge amplification. Paired-end libraries were sequenced from the clusters generated by an Illumina Hiseq X Ten platform (2 × 101 bp read length). Base calling was performed by the Illumina sequencer generated raw images through an integrated primary analysis software package called RTA (Real Time Analysis) and base calls binary was converted into FASTQ files using Illumina package bcl2fastq.

### 4.4. RNA-Seq Analysis

Sequences were trimmed for Illumina adapters and low-quality sequences using Trimmomatic v 0.38 [68]. To remove the Illumina TruSeq3 adapters, a code of ILLUMINACLIP:TruSeq3-PE-2.fa:2:30:10 was used. Quality encoding detected as Phred-33, number of threads 8, cut bases off the start and end of the read, if below a threshold quality both set at 3 to ensure that two consecutive bases had a score of 30 or more. The minimum length of the read to drop was specified at 36 bp and reads less than 36 bp were discarded.

Trimmed reads were mapped to the *Sorghum bicolor* v3.1.1 genome sequence using GSNAP (v 2018-03-25) (Wu et al., 2016) (-B 4 -N 1 -n 2 -Q -nofails format = sam). Genome assembly of *Sorghum bicolor* (v3.1.1) [69] was downloaded from Phytozome version 12.1. Samtools v 1.9 [70] was used to convert the raw SAM output from GSNAP to sorted BAM files. FPKM was calculated using sorted bam files with cufflinks (v2.2) [71]. Genes were classified as expressed if averaged FPKM (Fragments per Kilobase of transcript per Million mapped reads) surpass or equal to 1 in any time points [72]. HTSeq v 0.6.1 was used to extract the number of reads mapping to annotated exons of each gene for each RNA-seq library using union mode [73]. DESeq2 [74] was used to identify differentially expressed genes (DEGs) between resistant and susceptible lines, as well as the specified time points after SCA infestation. The criteria of log_2_ (fold change) ≥1 or ≤−1 and a probability value < 0.05 was used to declare DEGs.

### 4.5. Gene Ontology Enrichment and Clustering Analyses

Gene ontology (GO) and clustering analysis was performed on expressed genes (FPKM ≥ 1) from sorghum inbred lines, TAM428 and Tx2737, across time points of 0 h, 96 h, 216 h, and 360 h. Data of FPKM values were normalized by row and analyzed using the kmeans function implemented in R (v3.5.3) with 12 clusters. GO annotations were downloaded from phytozome (v 12.1) for sorghum (v 3.1.1). GO enrichment analyses of gene sets in each cluster were performed using GOATOOLS [75] with all annotated genes in the genome as background. GO terms were considered significantly enriched if *p* < 0.05 after controlling for false discovery rate using the Benjamini-Hochberg procedure. Then, singular enrichment analysis (SEA) for the DEG at 216 h and 360 h in the resistant genotype was conducted using agriGO v2 (GO analysis toolkit and database for the agricultural community [76]. Plant GO slim ontology type was selected from among the advanced options and used with the default statistical test method, multi-test adjustment method, and minimum number of mapping entries.

## 5. Conclusions

Sorghum infestation by SCA triggered a wide range of alterations in gene expression and cellular metabolism likely to impact plant growth and development. Sorghum responses to SCA infestation occurred quickly and were temporally dynamic. SCA infestation of the R genotype activated genes that regulate stress responses, cell communication, transferase activities, DNA binding, cellular catabolic processes, and various transcription factors. This study elucidates the gene groups in sorghum that respond to SCA feeding, and those that are either under- or over-expressed in a resistant line; these could be useful targets in breeding for improved germplasm resistance to SCA. However, further validation by qPCR may be of added value and needs to be carried out to improve the mechanistic or physiological insights into the biological system.

## Figures and Tables

**Figure 1 ijms-22-07129-f001:**
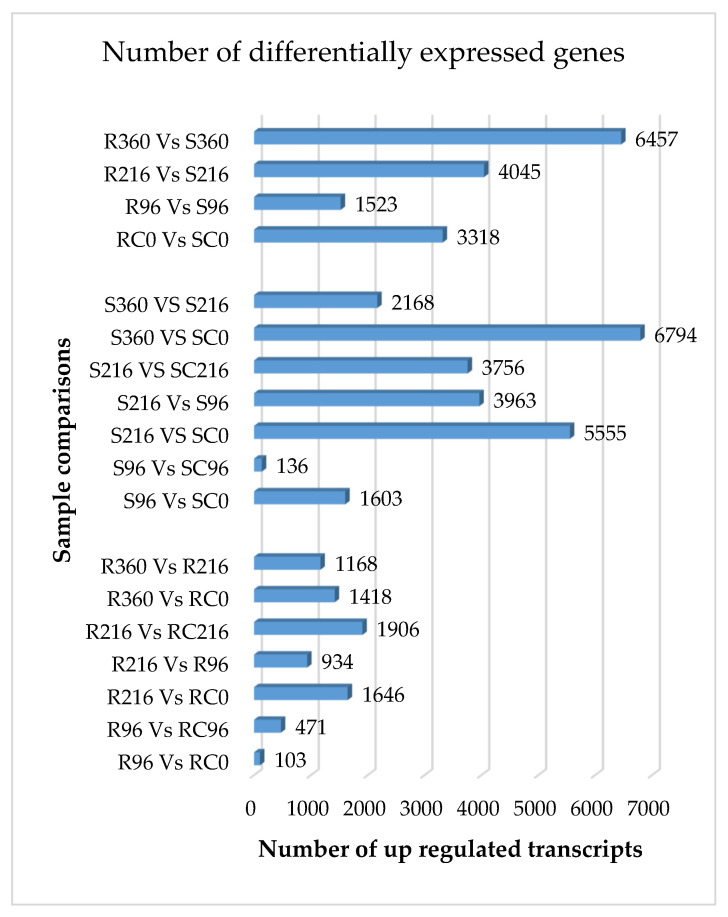
Number of DEGs (up regulated) in resistant and susceptible sorghum genotypes at different times after SCA infestation. Samples were taken 0 h, 96 h, 216 h, and 360 h after infestation of the plants with fourth instar apterous *Melanaphis sacchari*. R = resistant, RC = resistant control, S = susceptible, SC = susceptible control.

**Figure 2 ijms-22-07129-f002:**
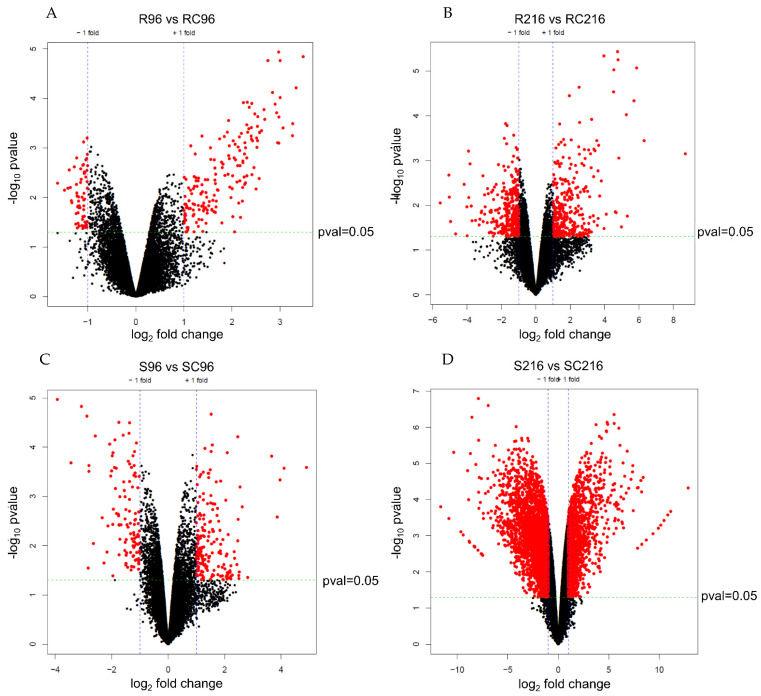
Volcano plots of selected differentially expressed genes in treatment and control plants of resistant and susceptible genotypes at selected time points after infestation. (**A**) Resistant control vs. resistant treatment at 96 h after infestation, (**B**) resistant control vs. resistant treatment at 216 h after infestation, (**C**) susceptible control vs. susceptible treatment at 96 h after infestation, and, (**D**) susceptible control vs. susceptible treatment at 216 h after infestation. Red dots indicate significance at *p* ≤ 0.05.

**Figure 3 ijms-22-07129-f003:**
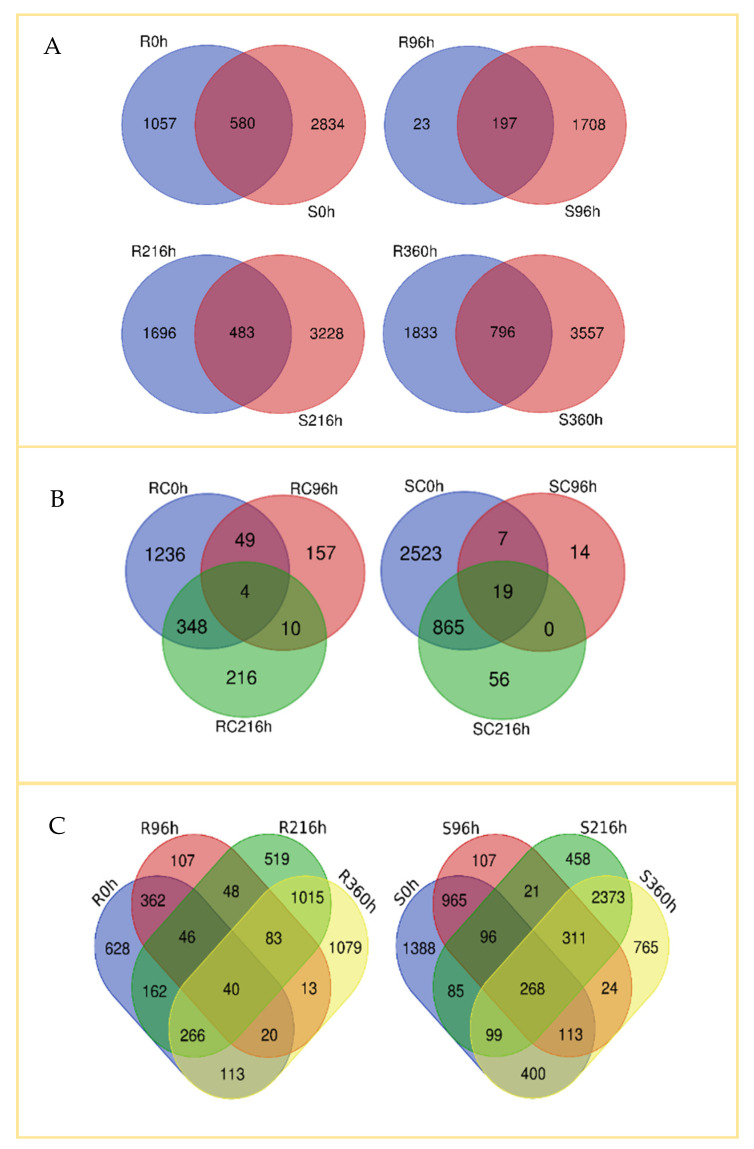
Venn diagram showing differences and commonalities in gene expression of resistant (TAM428, R) and susceptible (Tx2737, S) sorghum lines at different time points (0 h, 96 h, 216 h, and 360 h) after each plant was infested with five fourth instar nymphs of sugarcane aphid at the 3-leaf stage. Differentially expressed genes for each comparison were quantified by |log_2_fold changes| ≥1 and *p*-value < 0.05. Number of differentially and commonly expressed transcripts (**A**) between R and S sorghum lines at same time point, (**B**) between time points for untreated control, and (**C**) between time points for R and S lines post infestation.

**Figure 4 ijms-22-07129-f004:**
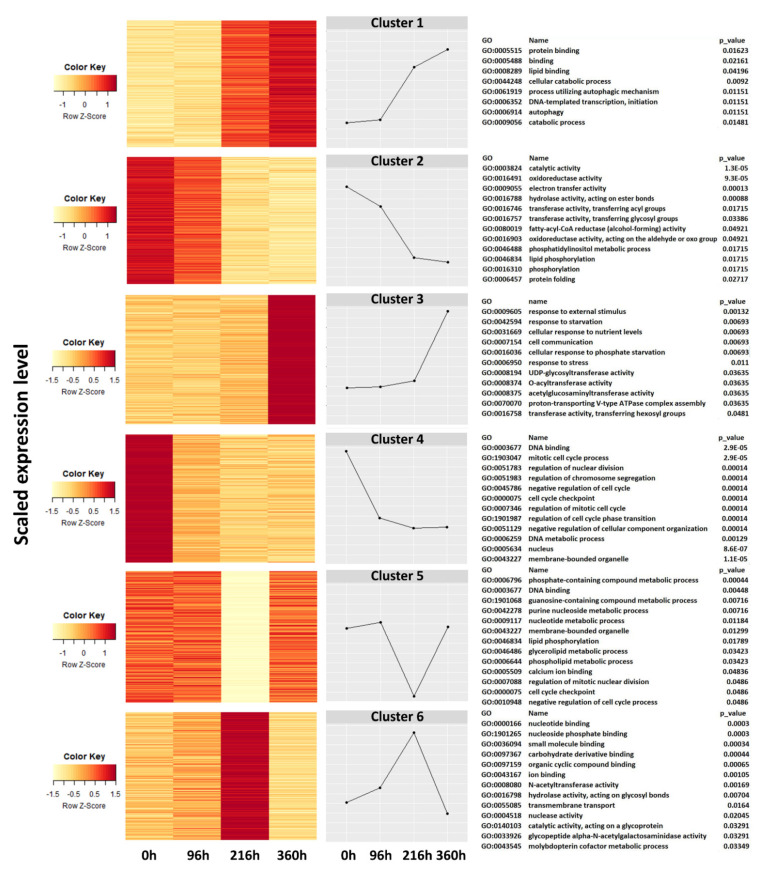
Heatmap and hierarchical clustering show gene expression pattern in resistant (TAM428) and susceptible (Tx2737) sorghum genotypes at different times post-infestation with sugarcane aphid. Each line in the figure represents a gene; color codes represent scaled gene expression level. Clustering analysis revealed six characteristic transcriptional changes in SCA resistant and susceptible genotypes.

**Figure 5 ijms-22-07129-f005:**
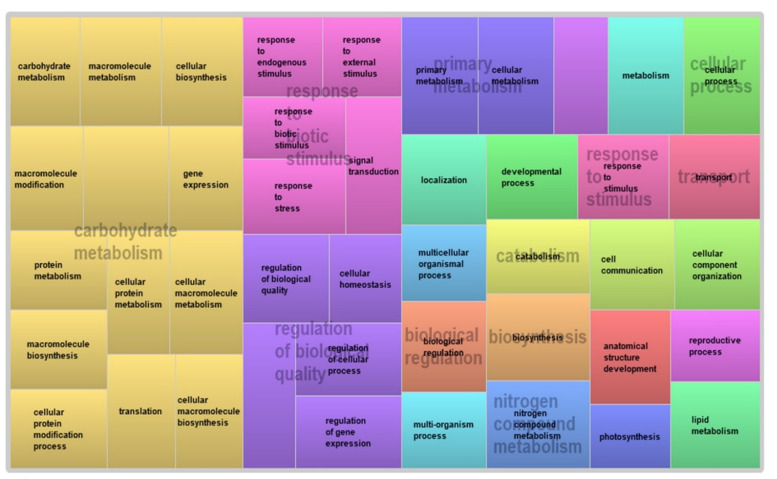
Annotation of DEGs in the R sorghum genotype at 360 h after sugarcane aphid infestation visualized with the reduced version of the plant gene ontology (REVIGO).

**Figure 6 ijms-22-07129-f006:**
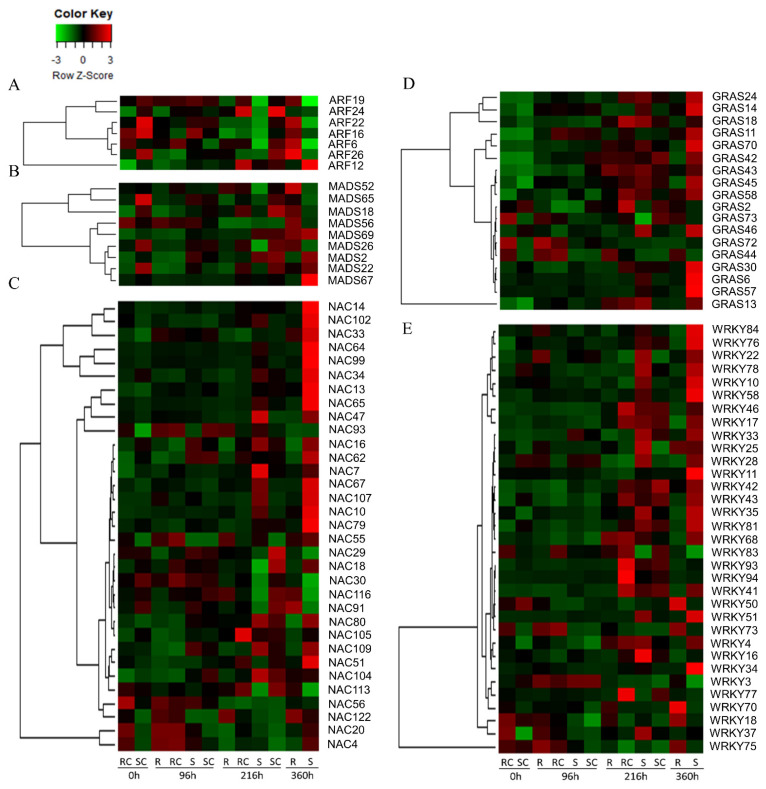
(**A**–**E**) Expression dynamics of transcription factors (TFs) in sugarcane aphid resistant and susceptible sorghum genotypes at different time points post-infestation. Different TF family members show dynamic expression profiles across genotypes and time points.

## Data Availability

All the raw sequencing reads for all the samples have been submitted to the National Centre for Biotechnology Information (NCBI) sequence read archive and deposited under the BioProject number PRJNA737745.

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
