# Peer review of "Comparative Transcriptome Analysis Reveals Genetic Mechanisms of Sugarcane Aphid Resistance in Grain Sorghum"

_ijms, 2021, doi:10.3390/ijms22137129_

Round 1

Reviewer 1 Report

Dear Authors,

I have a great honor and opportunity to review manuscript entitled: “Comparative transcriptome analysis reveals genetic mechanisms of sugarcane aphid resistance in grain sorghum” which is considered for publication in International Journal of Molecular Sciences. This work presents very interesting new insights in the case aphid resistance sorghum. This  article presents state of the art transcriptomic analyses and article is excellent written. However, I have some suggestions for this manuscript which I presents in a form of specific comments bellow.

Introduction section

I suggest to change in line 90 term “objectives” for “aims” it will fits better for article. I also suggest that aims will not presented after “:”. It could be transformed into :  “The present study was conducted to achieved three major aims. First was to characterize the genes that are 90 differentially expressed between SCA-resistant (TAM428) and susceptible (Tx2737) sorghum genotypes. Second was to elucidate patterns of temporal change in the expression of genes during SCA infestation. “ and so one for next aim.

Results section

Is very good written. In this part I suggest resize of some Figures because in some cases excellent results is difficult to observed on figures.  I suggest to enlarge Figure 2, Figure 3 Figure 5 and 6. Where is this possible I suggest to enlarge figures for whole page (especially for Figure 6). Moreover, I think that it will be better that Figure 5 should be enlarged and rotate to the vertical orientation for whole page.

Because of that reasons I recommend only minor revisions

Sincerely,

Author Response

Introduction section

Reviewer's comment:  I suggest to change in line 90 term “objectives” for “aims” it will fits better for article. I also suggest that aims will not be presented after “:”. It could be transformed into: “The present study was conducted to achieved three major aims. First was to characterize the genes that are 90 differentially expressed between SCA-resistant (TAM428) and susceptible (Tx2737) sorghum genotypes. Second was to elucidate patterns of temporal change in the expression of genes during SCA infestation. “and so one for next aim.

Answer: All changes made as suggested by the reviewer

 Results section

Reviewer's comment: Is very good written. In this part I suggest resize of some Figures because in some cases excellent results is difficult to observed on figures. I suggest to enlarge Figure 2, Figure 3 Figure 5 and 6. Where is this possible I suggest to enlarge figures for whole page (especially for Figure 6). Moreover, I think that it will be better that Figure 5 should be enlarged and rotate to the vertical orientation for whole page.

Answer: Edited the figures for clarity and visibility of the x and y axes labels. Appreciate the reviewer for the suggestion.

Reviewer 2 Report

The authors have performed the deep transcriptome analysis to identify the genetic mechanisms of sugarcane aphid resistance in grain sorghum. Authors have used RNA-Seq to conduct transcriptomics analysis on a moderately resistant genotype 19 (TAM428) and a susceptible genotype (Tx2737) to elucidate the molecular mechanisms underlying 20 resistances. Differential expression analysis revealed differences in transcriptomic profile between 21 the two genotypes at multiple time points after infestation by SCA. Authors have identified six gene clusters had differential 22 expression during SCA infestation. Genes, pathways and ontology classification have clearly demonstrated the mechanism of sugarcane aphid resistance in grain sorghum. Authors have well compiled the manuscript and presented with the very detailed analysis.  This is a well replicated study and has a great potential to get published in IJMS. Few comments to improve the manuscript,

  1. Figure 1 could be revised to show clearly up and down regulated genes. Also, x and y axis labels are not given. Y axis not showing the label for every bar, giving label for every bar could be easy for the understanding of the readers.
  2. Figure 2: Scale for the colors in the heat map is not provided.
  3. Figure 5 can be moved to the supplementary and also the figure can also be cropped to remove the web contents.
  4. A figure with putative scheme/pathway  of mechanism of resistance sugarcane aphid resistance in grain sorghum could be included based on the results.

Author Response

The authors have performed the deep transcriptome analysis to identify the genetic mechanisms of sugarcane aphid resistance in grain sorghum. Authors have used RNA-Seq to conduct transcriptomics analysis on a moderately resistant genotype (TAM428) and a susceptible genotype (Tx2737) to elucidate the molecular mechanisms underlying resistances. Differential expression analysis revealed differences in transcriptomic profile between the two genotypes at multiple time points after infestation by SCA. Authors have identified six gene clusters had differential expression during SCA infestation. Genes, pathways and ontology classification have clearly demonstrated the mechanism of sugarcane aphid resistance in grain sorghum. Authors have well compiled the manuscript and presented with the very detailed analysis.  This is a well replicated study and has a great potential to get published in IJMS. Few comments to improve the manuscript,

Reviewer's comment: Figure 1 could be revised to show clearly up and down regulated genes. Also, x and y axis labels are not given. Y axis not showing the label for every bar, giving label for every bar could be easy for the understanding of the readers.

Answer: Added x and y labels. The figure shows only the upregulated genes and the caption is corrected accordingly. Made all the comparisons visible now. We thank the reviewer for the comment.

Reviewer's comment: Figure 2: Scale for the colors in the heat map is not provided.

Answer: The color code was provided in the figure caption. Red dots indicate significance at p ≤ 0.05. Also included the scale for Figure 4, as heat map is Figure 4.

Reviewer's comment: Figure 5 can be moved to the supplementary and also the figure can also be cropped to remove the web contents.

Answer: Since we did not include the pathway analysis, we included Figure 5 to show the type and relative number of genes upregulated in the resistant genotype post infestation. Cropped and removed the web contents. If we move the figure to supplementary, the importance of those group of genes is overshadowed.

Reviewer's comment: A figure with putative scheme/pathway of mechanism of resistance sugarcane aphid resistance in grain sorghum could be included based on the results.

Answer: Refrained from including the pathway analysis as there are many genes involved and cannot give a clear pathway picture. Included rather Figure 5.

Reviewer 3 Report

The authors used NGS to evaluate transcript fluctuations in Sorghum in response to sugarcane aphid. This study produced a database of assembled and annotated transcriptome sequences useful as resource to reveal molecular mechanisms underlying the sugarcane aphid resistance in sorghum. They identified several genes involved in primary metabolism and in gene expression regulation.

Major comments:

This study is well done and well-presented and has generated a significant amount of data which will be very valuable to those working in this area. On the other hand, the results are largely descriptive and without further insight into plant mechanisms and processes, the paper will have limited broader interest. Moreover, RNA-seq data should be validated by RealTime qPCR assay.

Author Response

Reviewer's comments: The authors used NGS to evaluate transcript fluctuations in Sorghum in response to sugarcane aphid. This study produced a database of assembled and annotated transcriptome sequences useful as resource to reveal molecular mechanisms underlying the sugarcane aphid resistance in sorghum. They identified several genes involved in primary metabolism and in gene expression regulation.

This study is well done and well-presented and has generated a significant amount of data which will be very valuable to those working in this area. On the other hand, the results are largely descriptive and without further insight into plant mechanisms and processes, the paper will have limited broader interest. Moreover, RNA-seq data should be validated by RealTime qPCR assay.

Answer: Yes, we did not validate the RNA-seq data by RealTime qPCR assay. However, we presented the next generation sequencing data in this study and it didn't show any conflicts from the earlier studies' literatures for further validation/confirmation of results. We thank the reviewer for this critical comment.

Round 2

Reviewer 3 Report

In my opinion the author's reply is not satisfactory. RNA-seq data must be validated by Real Time PCR (at least part of the DEGs) on the same samples used for sequencing assay. Specially in studies, like is one, defining gene families or descriptive collection of transcripts and not providing mechanistic and/or physiological insights into the biological system or process being studied

Author Response

While appreciating the reviewer’s comment, we are justifying the following strongly for publication without performing RT-PCR:

Justification: We have performed RNA-Seq analysis in three replications. We have analyzed the data in a very conservative manner and based on the analysis of variance, the respective P values and the level of significance is presented for all the differentially expressed genes (DEGs) for the biotic stress resistance and some of them are known for insect resistance in the resistant genotype. These data were developed through the robust and highly reliable next generation sequencing (NGS) analysis in replications, which itself justifies validation for the RNA-Seq analysis. Following the same procedure, many earlier studies and some of the coauthors in this study published several RNA-Seq peer reviewed research articles on different stresses without complimentary validation using RTPCR.  

Round 3

Reviewer 3 Report

If RNA-seq methods and data analysis approaches are robust enough not always require validation by qPCR and/or other approaches, although there are situations where this may be of added value. In this study, the analytical pipeline is correct and the experimental design is well thought out, but it is a descriptive study without mechanistic or physiological insights into the biological system or process being studied. For this reason I suggested further insight on some interesting genes by real time PCR to enrich your manuscript. In my previous report I proposed to the editor to suggest you journals as Genes or similar for IF but I also told him that if he thought similar studies could be published on journal with an IMPACT FACTOR of 4.556 as IJMS he could proceed.

Author Response

We fully agree with the valuable comments by the reviewer. As we haven't done real time PCR for further validation in this study,  we added that point in the conclusion for the need of real time PCR to be carried out to add value for our findings to improve the mechanistic or physiological insights into the biological system.